# Effect of Fast Loading on the Seismic Performance of SRUHSC Frame Structures

**Wei Liu [1,2], Yingchao Ma [3,*] and Jinqing Jia [4]**

1   College of Hydraulic Science and Engineering, Zhengzhou University, Zhengzhou 450001, China; qiuniao1984@163.com
2   School of Ocean and Civil Engineering, Dalian Ocean University, Dalian 116023, China
3   Vanke Co., Ltd., Shenzhen 518085, China
4   Department of Construction Engineering, Dalian University of Technology, Dalian 116024, China; keyknown@126.com
*   Correspondence: tiger3969@163.com; Tel.: +86-15942604227

**Abstract:** Due to the high compressive strength and durability of ultra-high-strength concrete, SRUHSC (steel-reinforced ultra-high-strength concrete) frame structures have been used extensively in super-high-rise buildings. However, the SRUHSC showed obvious brittleness. Encasing structural steel in the material was recognized to be a good way of alleviating the problem of brittleness. The purpose of this study is to investigate the effect of the axial compression ratio on the seismic performance of a single-story, single-span SRUHSC frame structure under rapid loading. The failure mode, deformation, strength and stiffness degradation, energy dissipation capacity and residual displacement of the structure were compared and analyzed. The seismic performance of a single-story single-span SRUHSC frame structure is verified under the conditions of a fast loading rate and high axial compression ratio. The results suggest that the horizontal resistance capacity of structures can be significantly improved by fast loading in the elastic and elastic–plastic ranges. The ductility coefficient of the structure increases with the same axial compression ratio under fast loading. With an increase in loading rate, the secant stiffness of the structure is improved.

**Keywords:** axial compression ratio; ductility; horizontal resistance capacity; energy dissipation capacity; stiffness degradation

## 1. Introduction

In structural design, it is necessary to consider the seismic performance of a structure throughout its life. When the structure encounters earthquake action, the sensitivity rate of the material has an influence on the seismic performance of the whole structure.

At present, research on concrete structures mainly focuses on beams, columns, joints and other components under the action of rapid loading. Mahin et al. [1] reported that the flexural bearing capacity of beams under conditions of rapid loading was significantly improved, but the ductility and deformation capacity of the structure were not effectively improved. Chung et al. [2] studied 12 beams of different structures at different loading rates and found that the number of cracks was reduced in rapid loading, while the width of single cracks was increased. The damaged concrete protective layer was weakened but the ductility decreased. Kitajima et al. [3] conducted shaking-table tests on concrete columns and found that the damage from bi-directional loading was more serious than uni-directional loading, and the structural strength and stiffness degradation were significant. Inoue et al. [4] carried out static and one-way shaking-table loading tests on reinforced concrete columns and found that when the structure was subjected to vibration, the horizontal force of the structure was greater than the effect of static load. Lee et al. [5] found that with an increase in loading rate, the strength of the shear wall increased slightly, but the ductility performance decreased. Rodríguez-Martínez et al. [6] found that the bearing capacity of the

column also increased with an increase in loading rate. Fan and colleagues [7,8] studied the joints and side joints in concrete under different loading rates and different axial pressure ratios; they found that fewer cracks formed in the failure process, but the extension and expansion of a single crack were more serious than those under static loading. Bischoff and Perry [9] explored the effect of loading rate on concrete compression. The dynamic compressive strength of concrete increases with the increase in strain rate. The European standard CEB-FIP1990 [10] also pointed out that the dynamic strength of concrete increases with the increase in strain rate. Malvar and Ross [11] studied the effect of loading rate on the dynamic tensile strength of concrete. The experimental results showed that the dynamic tensile strength of concrete increased as the strain rate increased. Shafieifar et al. [12] determined the tensile and compressive behavior of UHPC (ultra-high-performance concrete) and a comparison is made with normal-strength concrete for the development of a numerical model to simulate the behavior of UHPC, using the finite element. The results obtained from the study indicated that the compressive strength of UHPC was three to four times greater than normal-strength concrete, and higher tensile strength and ductility of the material were observed, compared to regular concrete (two to four times greater). Maio et al. [13] proposed a diffuse cohesive interface approach to predict the structural response of UHPFRC (ultra-high-performance fiber-reinforced concrete) structures enhanced with embedded nanomaterials. The proposed fracture model was first validated by comparing the failure simulation results of UHPFRC specimens containing different fractions of graphite nanoplatelets with the available experimental data.

In conclusion, the global research on reinforced concrete structures is still at the component level, while the response of steel ultra-high-strength concrete frame structures under the action of a rapid load is an unknown field. The following problems need to be addressed:

(1) In the study of fast loading, the existing building standards lag seriously behind the engineering. The European code CEB-FIP1990 proposed that the strength of concrete increases with an increase in the fast-loading rate. However, there is no relevant code for fast loading in China, which is problematic for further study of the seismic performance of structures.

(2) The existing studies on the strain rates of high-strength and ultra-high-strength concrete materials mostly focus on the material level. The seismic performance analysis of materials when they are incorporated into structures is still unknown in terms of understanding the impact of fast loading on the overall structure.

(3) At present, most of the research studies are based in the laboratory, and most of them are on reinforced ordinary concrete beams, columns, joints and other components. Research on overall rapid loading rates and frame structure is yet to be conducted.

Based on the above situation, this paper focuses on a study of the effect of the axial compression ratio on the seismic performance of a single-story, single-span steel frame structure with ultra-high-strength concrete under rapid loading. The failure mode, deformation, strength and stiffness degradation, energy dissipation capacity and residual displacement of the structure are compared and analyzed. The seismic performance of the single-story, single-span SRUHSC frame structure is verified under the conditions of a high loading rate and high axial compression ratio.

## 2. Experimental Program

### 2.1. Design of Specimens

The systems detailed in Table 1 consist of four trusses of single-layer and single-span steel frame ultrahigh-strength concrete frame structures. The main parameters considered in the study included the section size of the beam, specific section structure form, shear span ratio of the section form of the frame column, test axial compression ratio ($n_t$) and loading rate. The design is based on the principle of "strong column, weak beam; strong node, weak component".

**Table 1.** Design parameters of the specimens.

| Specimens | Steel Shape | Structural Steel Ratio | Sectional Strength Ratio | Shear Span Ratio | Test Axial Compression Ratio | Loading Rate (mm/s) |
|---|---|---|---|---|---|---|
| SRUHSC-SRC-N25-I3 | I10 | 3.5% | 1.4 | 3 | 0.25 | 0.5 |
| SRUHSC-SRC-N45-I3 | I10 | 3.5% | 1.4 | 3 | 0.45 | 0.5 |
| SRUHSC-SRC-N25-I3-V20 | I10 | 3.5% | 1.4 | 3 | 0.25 | 20 |
| SRUHSC-SRC-N45-I3-V20 | I10 | 3.5% | 1.4 | 3 | 0.45 | 20 |

Note: SRUHSC stands for steel-reinforced ultra-high-strength concrete; SRC stands for steel-reinforced concrete; N is the axial compression ratio; I represents the steel bone forms; 3 represents the shear span ratio; V is the loading rate.

According to the building code for the seismic design of buildings, the design's beam–column section strength ratio is 1.4. The total height of the column is 1500 mm. The slenderness ratio is 7.5, which meets the requirement of less than 8 specified in the code and avoids the lateral buckling phenomenon of the column during the loading process. The calculated height of the column is 1200 mm and 800 mm, and the slenderness ratio is 6 and 4, respectively. The beam-column linear stiffness ratio is less than or equal to 0.45, which meets the overall seismic requirements of the frame structure. The frame structure with a shear-span ratio of 3 is displayed, as shown in Figure 1a.

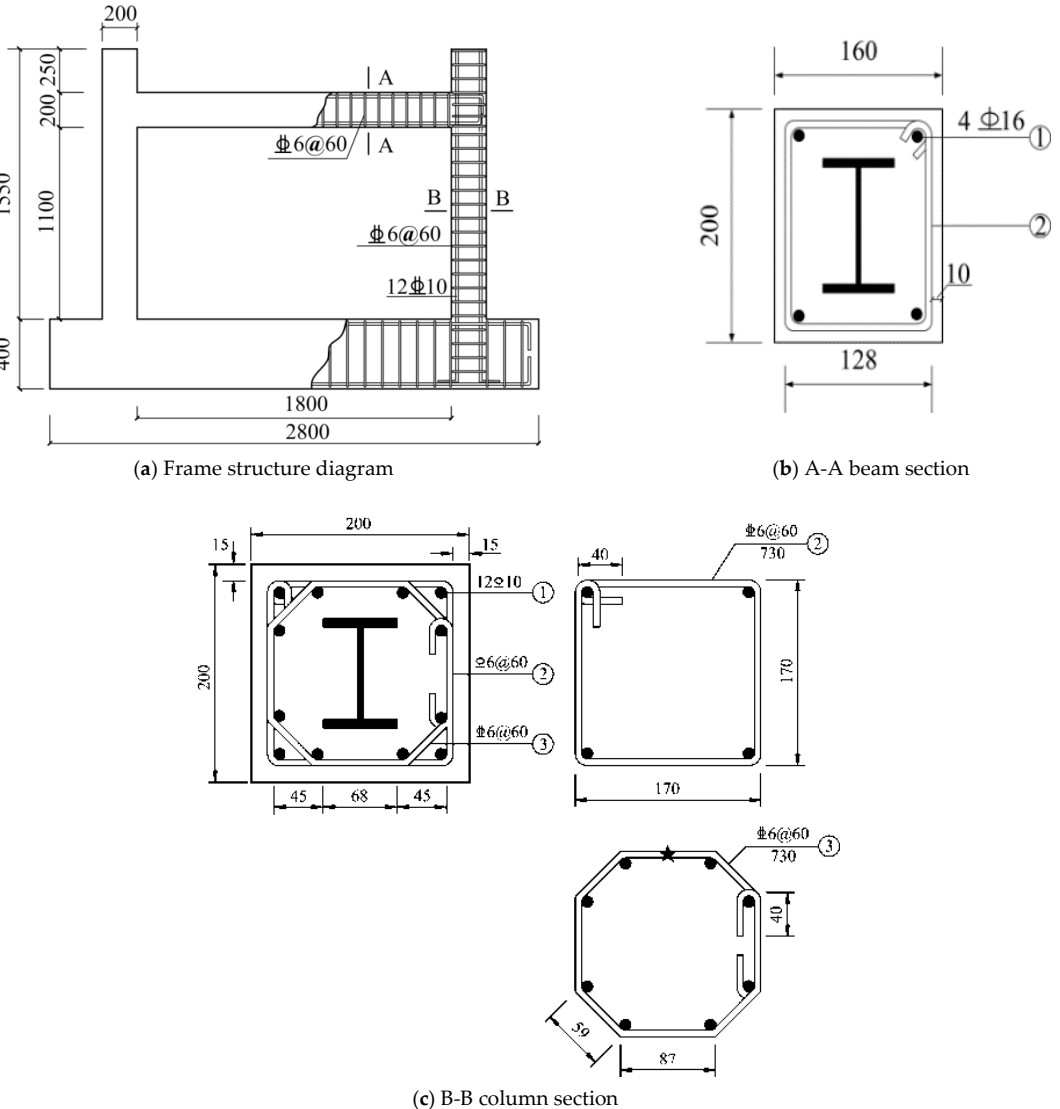

(**a**) Frame structure diagram

(**b**) A-A beam section

(**c**) B-B column section

**Figure 1.** Geometry and reinforcement details of test specimens (units: mm).

The compressive strength of the column is C100 and that of the beam is C40. The mechanical properties are shown in Tables 2 and 3. The beam section size is 160 mm × 200 mm, while the stirrup is a rectangular hoop of HRB400 ⏇ 6, as shown in Figure 1b. The section size of the column is 200 mm × 200 mm. The longitudinal reinforcement is of HRB400 three-level rebar 12 ⏇ 10. The stirrup is the composite stirrup of HRB400 ⏇ 6. The stirrup spacing is 60 mm and the longitudinal reinforcement of the beam is HRB335 4 ⏇ 16. The cross-section form of the column is shown in Figure 1c. The construction style of the SRUHSC frame is shown in Figure 2.

**Table 2.** Mechanical properties of concrete.

| Concrete Strength | Cube Crushing Strength/$f_{cu}$ (MPa) | Prismatic Compressive Strength/$f_c$ (MPa) | Elasticity Modulus/Ec (GPa) | Poisson Ratio/ν |
|---|---|---|---|---|
| | 116.57 | 104.39 | 43.27 | 0.241 |
| C100 | 115.36 | 108.45 | 44.25 | 0.248 |
| | 108.72 | 103.72 | 45.32 | 0.246 |
| Average value | 113.55 | 105.52 | 44.28 | 0.245 |
| | 46.26 | 41.57 | 34.01 | 0.222 |
| C40 | 48.76 | 40.82 | 32.96 | 0.220 |
| | 46.88 | 42.26 | 32.48 | 0.206 |
| Average value | 47.30 | 41.55 | 33.15 | 0.216 |

**Table 3.** Mechanical properties of reinforcement and structural steel.

| Mechanical Performance Index | φ6 (HRB400) | φ10 (HRB400) | φ16 (HRB335) | I10 (Q235) | |
|---|---|---|---|---|---|
| | | | | Flange | Web |
| Yield strength $\bar{f}_y$ (MPa) | 522.7 | 437.0 | 383.0 | 317.2 | 305.2 |
| Yield strain $\bar{\varepsilon}_y$ ($\times 10^{-6}$) | 2736 | 2309 | 1888 | 1540 | 1502 |
| Ultimate strength $\bar{f}_u$ (MPa) | 680.2 | 616.8 | 579.6 | 424.8 | 394.9 |

Note: The mass density of steel, shear modulus and coefficient of linear expansion are ρ = 7850 kg/m³, G = 79 × 10³ MPa and α = 12 × 10⁻⁶/°C, respectively.

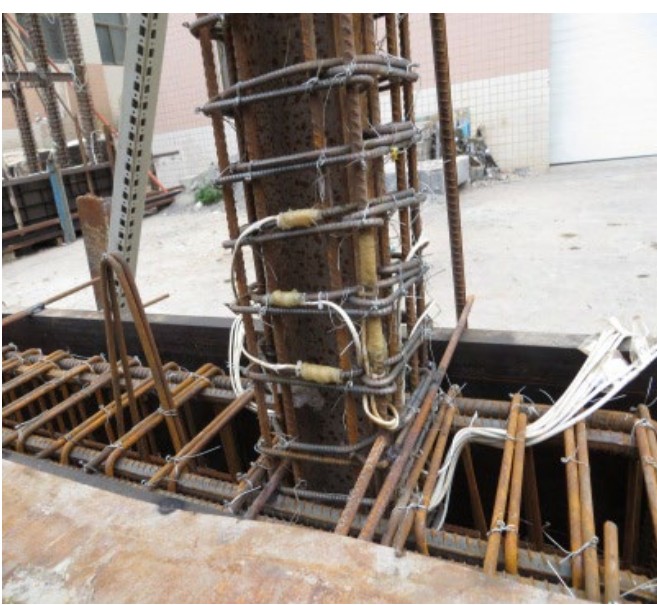

**Figure 2.** The construction details of an SRUHSC frame.

### 2.2. Loading Conditions

The test setup for all specimens is shown in Figure 3. The loading spectrum is shown in Figure 4. The first three amplitudes are cycled once at each stage, while the increment of the displacement angle is 0.25%. Starting from the fourth cycle, the amplitude is cycled three times at each stage and the increment of the displacement angle is then 0.5%. The test is completed when the structure loses its resistance capacity or the horizontal peak load drops to 80% of the peak load. The seismic performance of the frame structure is analyzed using the fast-loading method. The size of the frame structure's loading rate and the structure's dynamic characteristics have a close relationship with seismic intensity. Usually, the loading rate remains the same in the process of an earthquake. Hence, the seismic spectrum delta maximum displacement and displacement velocity can be calculated. The time to reach the maximum displacement is calculated as follows [9]:

$$T_{max} = \frac{\Delta_{max}}{V} \tag{1}$$

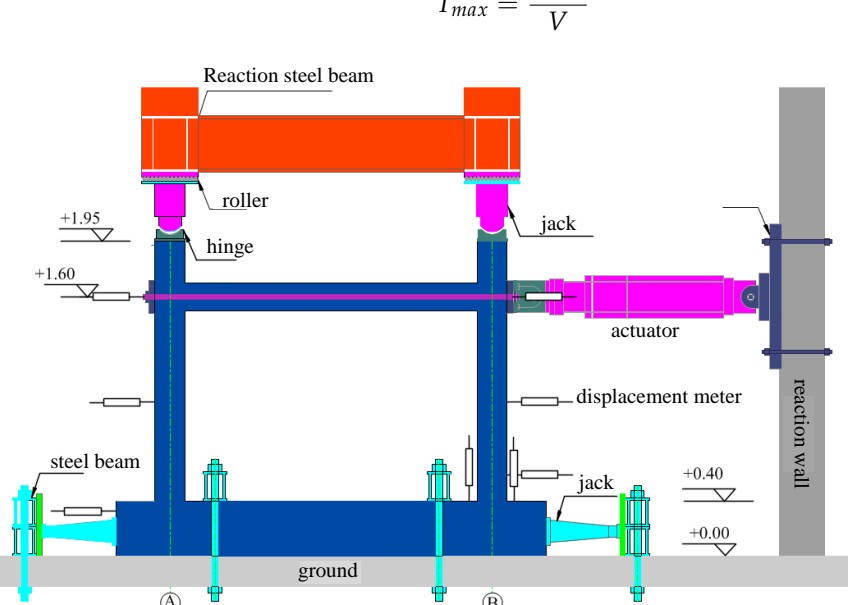

**Figure 3.** Test setup for all specimens.

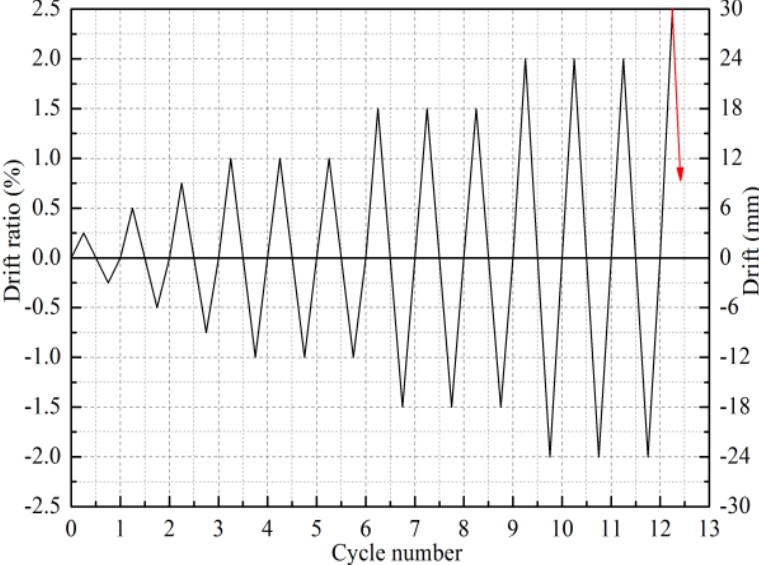

**Figure 4.** Loading spectrum.

If the structure reaches the yield state during this displacement cycle, the strain rate of the structure can be calculated from the yield strain:

$$\varepsilon = \frac{\varepsilon_y}{T_{max}} \tag{2}$$

The essential parameter of these loading conditions is the loading rate. The loading rates of this test are 0.5 mm/s and 20 mm/s, respectively, and the corresponding material strain rates are $9.6 \times 10^{-5}$ and $3.8 \times 10^{-3}$, respectively. For a reinforced concrete structure, the loading rate of 2 mm/s is a critical value. When the loading value is larger than this value, the structure is affected by the loading rate [14,15]. The concrete stress increases significantly with the increase of the loading rate. At the same time, the reinforcement also has a certain rate sensitivity under dynamic load, while its stress-strain relationship is related to the loading rate, to a certain extent. However, the elastic modulus and ultimate strength of the reinforcement are not significantly affected by the loading rate [16–21].

In the fast—loading process, the structure needs to respond quickly to changes in the loading direction at the peak point of each cycle. The speed and acceleration change direction instantly. However, due to the inertia of the structure, it is impossible for the loading device to meet the loading conditions simultaneously. Hence, there will be a tiny transition section at the peak displacement of the structure that can meet the transformation of the structure acceleration. Its displacement is slightly less than the loading spectrum amplitude. The impact of the velocity and acceleration on the overall bearing capacity and stiffness of the structure can also be observed.

*2.3. Crack Pattern and Failure Mode*

In the rapid-loading process, the failure process and crack development of the whole structure were visibly different from that of the pseudo-static test, due to the fast deformation rate of the whole structure. Firstly, the number of cracks at the end of the beam and the column foot was less than that in the pseudo-static test. However, the speed of cracks in the beam end and the column foot was faster in the rapid-loading test because the cracks in the pseudo-static test developed along the interface between the aggregate and cladding slurry, which has a certain irregularity. In the process of rapid loading, cracks are formed in a certain direction due to the fast speed of crack formation. The cracks occur at the interface between aggregates, as well as cladding slurry at the coarse aggregate section. Therefore, small cracks appeared in the quasi-static test. The number of structural cracks in the rapid-loading test was relatively small.

In the specimens where $n_t = 0.25$, the crack formation at the end of the beam and the speed of crack opening were both quick. When the beam was loaded at a displacement angle of 0.5%, small vertical cracks appeared at the upper left end and the lower right end of the beam at the same time. When the loading continued to a 1% displacement angle, the number of cracks in the upper left end and lower right end of the beam increased significantly. At this time, cracks in the protective concrete layer began to appear within the range of 30 mm, accompanied by an audible sound. The concrete cracks widened in the range of 100 mm at the beam end, and small oblique cracks appeared in the core area of the joints and crushed the protective concrete layer in the local area of the column foot. When the displacement angle reached 3%, the protective concrete layer fell off on both sides of the column foot. With further loading at a 3.5% displacement angle, the concrete was crushed and fell off within 100 mm of the beam end, and oblique cracks appeared in a 45° direction from the node core area. When the displacement angle reached 4%, the concrete at the beam end was crushed and the vertical cracks in the concrete at the bottom of the column increased. At this point, the test-bearing capacity had reached 80% of the ultimate bearing capacity. The destruction of the interface between each component was more serious. Cracks at the bottom of the column were mainly composed of horizontal cracks, while vertical cracks were mainly focused on the edges of the concrete. Due to the brittle property of ultra-high-strength concrete itself, the damage in the area of the column

foot was more serious than that in the pseudo-static test. The damage at the bottom of the column was mainly concentrated in the height range of the double section, which was concentrated and intense, as shown in Figure 5a,b.

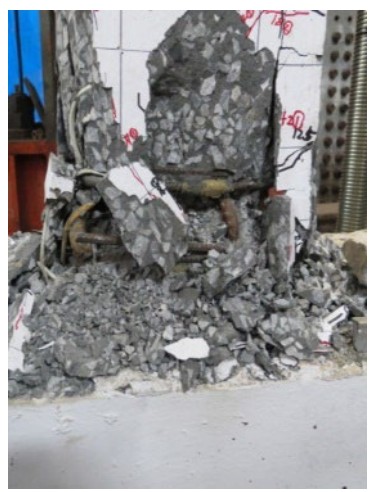

(**a**) $n_t$ = 0.25, 0.5 mm/s (pseudo-static)

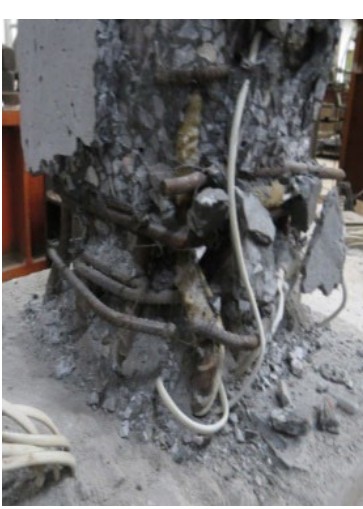

(**b**) $n_t$ = 0.25, 20 mm/s (fast loading)

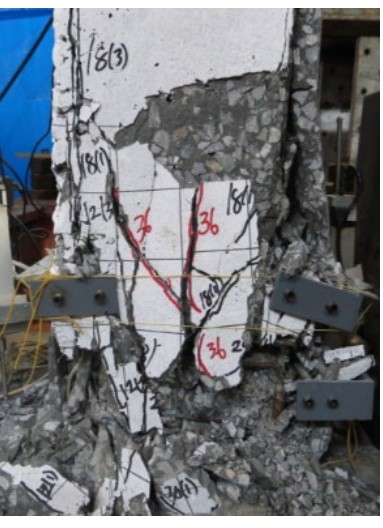

(**c**) $n_t$ = 0.45 0.5 mm/s (pseudo-static)

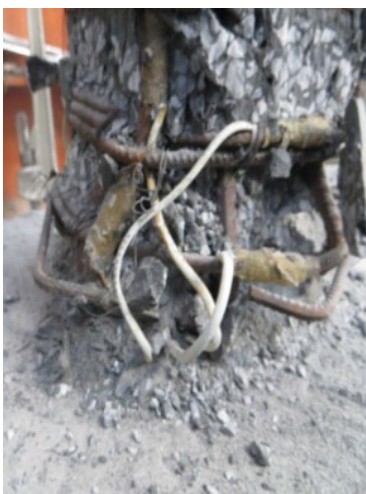

(**d**) $n_t$ = 0.45 20 mm/s (fast loading)

**Figure 5.** The cracks at the bottom of the column.

In the specimens with $n_t$ = 0.45, the destruction of the structures occurred more quickly. The first vertical cracks formed at the bottom of the column. Compared with the pseudo-static test, no obvious oblique cracks formed at the bottom of the column, but the destruction occurred more quickly. The damage in the core area was more severe. When the structures were finally destroyed, the longitudinal bars at the bottom of the column and steel bones were buckled and the stirrups were pulled out and broken. The destruction was more serious than in the static test, as shown in Figure 5c,d.

For the specimens with $n_t$ = 0.25 and $n_t$ = 0.45, the increase in the axial compression ratio weakened the beam end and accelerated the failure of the column bottom when the loading rate reached 20 mm/s. The vertical crack extension speed was accelerated, and the protective concrete layer collapsed, indicating that a large amount of energy was accumulated in the SRUHSC frame structure under the conditions of a high axial compression ratio and rapid loading. When the integrity of the structural section changed, the energy in the structure was released instantly, resulting in concrete collapse, as shown in Figure 5b,d.

## 3. Results and Discussion

### 3.1. Load versus Displacement Hysteresis Loops

In tests with the same axial compression ratio and different loading rates, there are certain differences between the load versus displacement hysteresis loops. In the specimens with $n_t$ = 0.25, the load versus displacement hysteresis loops were in an approximately straight line at the initial loading stage, which is similar to the change of frame structure at a low rate. As the loading continued, the horizontal displacement value increased and the failure of the structure was aggravated at a high rate. The peak force of the hysteresis loop was higher than that of the pseudo-static test at the same amplitude. The values of fast loading were similar to those of pseudo-static loading after yielding. The peak value of fast loading was slightly higher than that of pseudo-static loading. The influence of different loading rates on the same frame structure is not obvious. It is clear from the hysteresis loops that the bond-slip between steel and reinforced concrete has led to the plastic deformation stages under the amplitude at each level. However, the horizontal displacement of the structure increased under rapid loading and the slope of horizontal force decline is significantly smaller than that of the pseudo-static test in the hysteresis loop bond-slip section, as shown in Figure 6a.

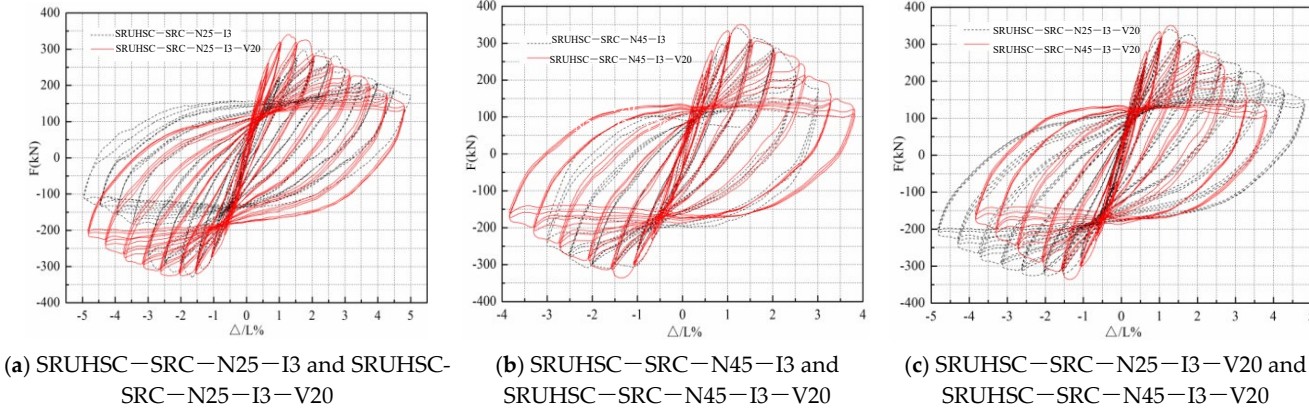

(**a**) SRUHSC−SRC−N25−I3 and SRUHSC-SRC−N25−I3−V20

(**b**) SRUHSC−SRC−N45−I3 and SRUHSC−SRC−N45−I3−V20

(**c**) SRUHSC−SRC−N25−I3−V20 and SRUHSC−SRC−N45−I3−V20

**Figure 6.** Comparison of the hysteresis loops.

For the specimens with $n_t$ = 0.45, the peak force of each amplitude of rapid loading is obviously greater than that of pseudo-static loading before the peak load is reached. After peak load, the shapes of the two hysteresis loops are extremely similar and the peak force is also similar. Before collapsing, the displacement angle of rapid loading is greater than that of the pseudo-static test and the deformation capacity of the structure is improved, as shown in Figure 6b.

The horizontal force of the frame structure was significantly increased by rapid loading before it reached the peak load under a normal axial compression ratio and high axial compression ratio. However, the horizontal force of the structure at different rates was similar after the structure entered the plastic deformation stage completely. It can be seen that the loading rate has a significant effect on the elastic and elastic-plastic stages of the structure. The effect of the loading rate on the horizontal bearing capacity of the structure was weakened after plastic deformation was achieved. At the same time, the loading rate reduced the decreasing rate of the horizontal bearing capacity of the structure during the plastic deformation stage. A certain effect takes place on the stability of the structure and the increase in the deformation capacity of the structure.

In the two frames with different axial compression ratios at high loading rates, the increase in axial compression ratio had no significant effect on the peak load of the structures. However, the failure of the concrete at the beam end and column foot intensifies after the peak load. The failure of bond slips among the steel bar, steel bone and concrete increases. The slope of decline in the horizontal force increases and the horizontal deformation

capacity of the structure decreases. Its variation trend is consistent with that of the structure under the quasi-static test.

### 3.2. Skeleton Curves

For the specimens with $n_t = 0.25$, the slope of the bearing capacity of the fast-loading structure on the push side is greater than that of the quasi-static test, as shown in Figure 7a. In reverse loading, the slope of the horizontal force rising is the same before the structure reaches the peak load. In the descending section, the slope of a fast-loading structure is smaller than that of the static stage. The main reason is that the serious damage seen at the beam end is related to the damage imbalance in the process of pushing and pulling under rapid loading. By averaging the positive and negative peak loads, it is apparent that the horizontal load of the structure is greater than the peak load of the pseudo-static frame under the axial pressure ratio. Therefore, it was verified that the increase in loading rate enhanced the strength of the structure and improved the lateral resistance of the structure.

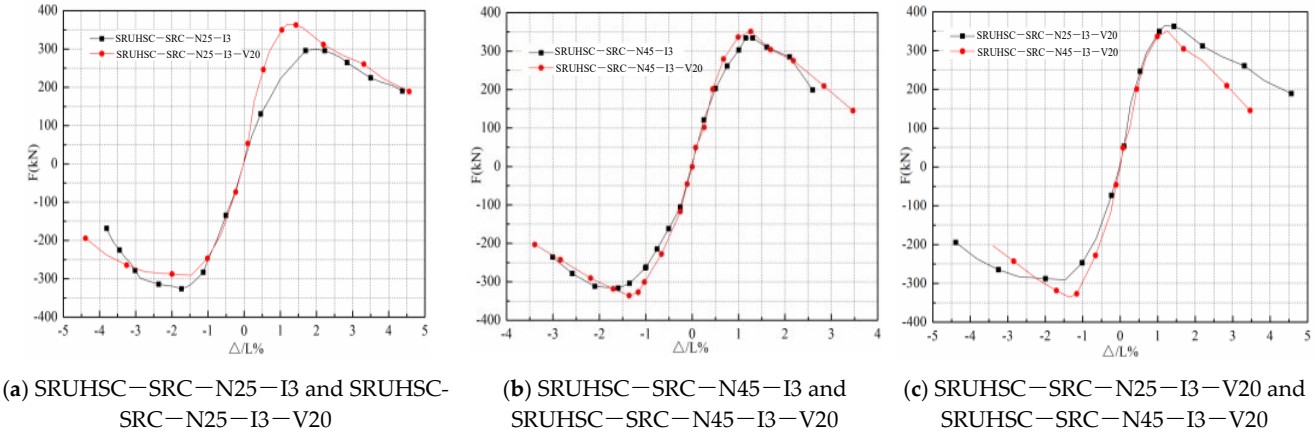

(**a**) SRUHSC−SRC−N25−I3 and SRUHSC-SRC−N25−I3−V20

(**b**) SRUHSC−SRC−N45−I3 and SRUHSC−SRC−N45−I3−V20

(**c**) SRUHSC−SRC−N25−I3−V20 and SRUHSC−SRC−N45−I3−V20

**Figure 7.** Comparison of skeleton curves.

When the axial compression ratio was 0.45, the variation trends of the two frame structures were very close, as shown in Figure 7b. The peak load of the frame under high-speed loading was slightly larger than that of the quasi-static structure. The variation trend of the structure is consistent. However, the slope of the structure under fast loading is large in the descending section of the structure. The effect of the loading rate on a structure with a high axial compression ratio was not observed.

In Figure 7c, the changes in the two curves are very clear when under the conditions of the same loading rate and a different coaxial pressure ratio. The slope of the bearing capacity rise and the decline of a frame structure with a high axial pressure ratio were both very steep, while the displacement angle decreased. It was verified that the slope of the load increased with the increase in the axial compression ratio under the condition of a high loading rate, which is consistent with the phenomenon of the law of quasi-static loading tests.

### 3.3. Ductility

To compare the ultimate deformation capacity of the frame structure, the inelastic deformation was quantified by displacement ductility, $\mu_\Delta$. Based on the characteristic hysteretic responses of the frame structure, the displacement ductility $\mu_\Delta$ is defined as follows [16,17]:

$$u_\Delta = \frac{\Delta_y}{\Delta_u} \tag{3}$$

where $\Delta_u$ and $\Delta_y$ are the ultimate and yielding displacement, respectively. $\Delta_u$ is defined as the post-spalling displacement, where the residual lateral force has declined to 85% of $F_{max}$, which depends on the final failure mode.

It can be seen from Table 4 that the displacement ductility coefficient of the frame structure was relatively small. The interlayer displacement angle, corresponding to the ultimate displacement angle, was between 1/25 and 1/48, which meets the specific requirements of the specification that it should be greater than 1/50. In addition, the horizontal force was still unequal under positive and negative loading, which was the same as that under quasi-static conditions. During the process of structure failure, the cracking was less marked but the damage from each of the single cracks was more serious.

**Table 4.** Skeleton curve eigenvalue.

| Specimens | | Yield Point | | Peak Point | | Ultimate Point | | | | Ductility |
|---|---|---|---|---|---|---|---|---|---|---|
| | | Fy (kN) | Δy/mm | Fm (kN) | Δm (mm) | Fu (kN) | Δu/mm | θ_u | μ_Δ | Average Value |
| SRUHSC-SRC-N25-I3 | + | 300.2 | 15.38 | 326.22 | 20.56 | 261 | 37.47 | 1/31 | 2.44 | 2.39 |
| | − | −280.7 | −16.72 | −300.38 | −23.26 | −240.3 | −39.1 | 1/32 | 2.34 | |
| SRUHSC-SRC-N25-I3-V20 | + | 303.9 | 10.6 | 340 | 15.67 | 270 | 30.7 | 1/39 | 2.89 | 3.29 |
| | − | −285.3 | −13.3 | −321 | −18.46 | −256 | −49.1 | 1/25 | 3.69 | |
| SRUHSC-SRC-N45-I3 | + | 292.6 | 11.34 | 332.6 | 13.89 | 266.1 | 27.3 | 1/44 | 2.41 | 2.44 |
| | − | −279.7 | −13.7 | −318 | −19.15 | −254.4 | −33.86 | 1/35 | 2.47 | |
| SRUHSC-SRC-N45-I3-V20 | + | 321 | 10.1 | 351.3 | 15.36 | 281 | 25.1 | 1/48 | 2.49 | 2.64 |
| | − | −291.4 | −11.3 | −337.1 | −16.19 | −269.68 | −31.5 | 1/38 | 2.78 | |

Note: $\theta_u$ is the ultimate displacement angle.

### 3.4. Stiffness Degradation

Stiffness degradation is an important parameter of seismic performance, which is used to evaluate the degree of structural damage under seismic load. This paper has mainly studied the influence of different loading rates and different axial compression ratios on the stiffness degradation of steel-bone ultra-high-strength concrete frame structures in the process of rapid loading.

When the axial compression ratio is 0.25, the initial stiffness of the structure under rapid loading is significantly higher than that of the pseudo-static test. When the displacement angle is 1%, the frame structure is near the yield point, which is an inflection point of the structure change. When the displacement angle is less than 1%, the secant stiffness of the structure is significantly higher than that of the pseudo-static test component, and the stiffness degradation slope is very high. As shown, the degree of damage degree in the structure was more serious under rapid loading. The stiffness declined rapidly; when the displacement angle was more than 1%, the fast-loading frame stiffness degradation slope was reduced. Along with the increase in the displacement angle, the slope of the degradation gradually equaled the slope approximation of the quasi-static structure degradation. It can be seen that the influence of fast loading on the structural stiffness degradation was weakened at the stage of plastic deformation, as shown in Figure 8a.

When the axial compression ratio was 0.45, the fast-loading framework of secant stiffness values was greater than in the pseudo-static test load with the same displacement angle. The overall stiffness was higher than that of the quasi-static frame structure but the stiffness degradation declined. This shows that fast-loading can increase the stiffness of the structure as a whole, under a high axial compression ratio, but it has no effect on the secant stiffness degradation, as shown in Figure 8b.

As shown in Figure 8c, the secant stiffness value and stiffness degradation slope of the structure increased with the increase in the axial compression ratio. The displacement angle of the overall deformation of the structure was smaller, which is consistent with the law of the pseudo-static test [18–22]. As shown, under the action of a high loading rate, the stiffness degradation rule was consistent with the quasi-static test as the axial compression ratio changed.

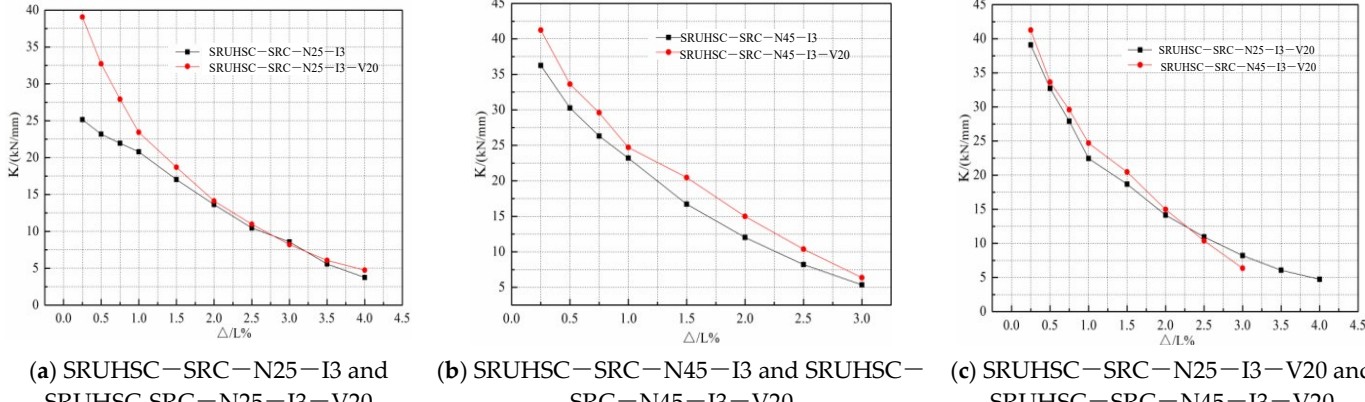

(**a**) SRUHSC－SRC－N25－I3 and SRUHSC-SRC－N25－I3－V20

(**b**) SRUHSC－SRC－N45－I3 and SRUHSC－SRC－N45－I3－V20

(**c**) SRUHSC－SRC－N25－I3－V20 and SRUHSC－SRC－N45－I3－V20

**Figure 8.** Comparison of stiffness degradations.

### 3.5. Energy Dissipation

The energy dissipation capacity of each structure was close to the others at the initial stage of loading. Under the action of continued loading, the energy dissipation capacity of the fast-loading frame was smaller than that of the static frame. When the axial compression ratio was 0.25 and the displacement angle was 2.5%, the energy dissipation capacity of the two structures was significantly different. The energy dissipation capacity of the fast-loading structure was smaller than that of the quasi-static structure, as shown in Figure 9a. When the displacement angle of the structure exceeded 1%, the energy dissipation capacity of the two structures with a 0.45 axial compression ratio was significantly different, as shown in Figure 9b. The energy dissipation capacity of the fast-loading frame was smaller than that of the quasi-static frame structure. It was shown that the rapid loading had little effect on the energy dissipation capacity of the whole frame structure under a normal axial compression ratio. The energy dissipation capacity of the structure under rapid loading was lower than that of the quasi-static test. For the frame with a high axial compression ratio, the energy dissipation capacity of the fast-loading frame was lower than that of the static structure after yielding. In conclusion, the influence of fast loading on the energy dissipation capacity of the structure is not obvious, and the energy dissipation capacity is slightly lower than the test value of static loading.

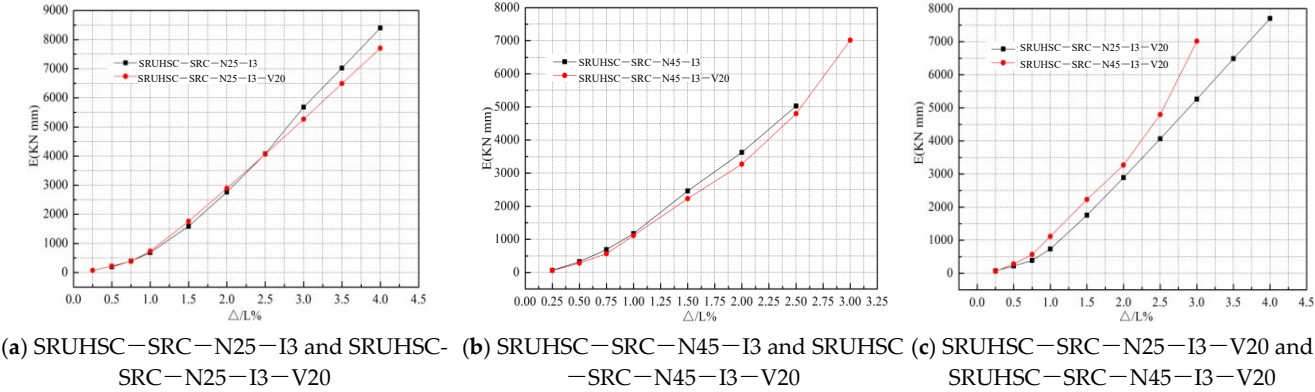

(**a**) SRUHSC－SRC－N25－I3 and SRUHSC-SRC－N25－I3－V20

(**b**) SRUHSC－SRC－N45－I3 and SRUHSC－SRC－N45－I3－V20

(**c**) SRUHSC－SRC－N25－I3－V20 and SRUHSC－SRC－N45－I3－V20

**Figure 9.** Dissipation capacity of fast loading.

At the same loading rate (20 mm/s) and the same displacement angle, the energy dissipation capacity of the structure increased with the increase in the axial compression ratio. In terms of the energy dissipation capacity of the structure, the influence of the change of axial compression ratio on the energy absorption of the structure was consistent with that of the pseudo-static test, indicating that the axial compression ratio has no evident effect on the energy dissipation capacity of the structure under fast loading.

### 3.6. Residual Displacement

The residual displacement ratio is the ratio of the residual displacement of the structure to the corresponding amplitude, which can more directly reflect the changes of the residual displacement of the structure under the action of horizontal load. The residual displacement of the structure is very small, within the 1% displacement angle. There are obvious differences between the residual displacements of the structure when the displacement angle reached 1%, as shown in Figure 10. When the axial compression ratio is 0.25, the residual displacement of the quasi-static structure is significantly greater than the residual displacement under rapid loading. Due to the material rate correlation, the strength and deformation capacity of steel is improved to a certain extent. The structure of the recovery ability is enhanced. The residual displacement of the structure is much smaller than that of the static structure, which is beneficial in terms of the overall stability and seismic performance of the structure.

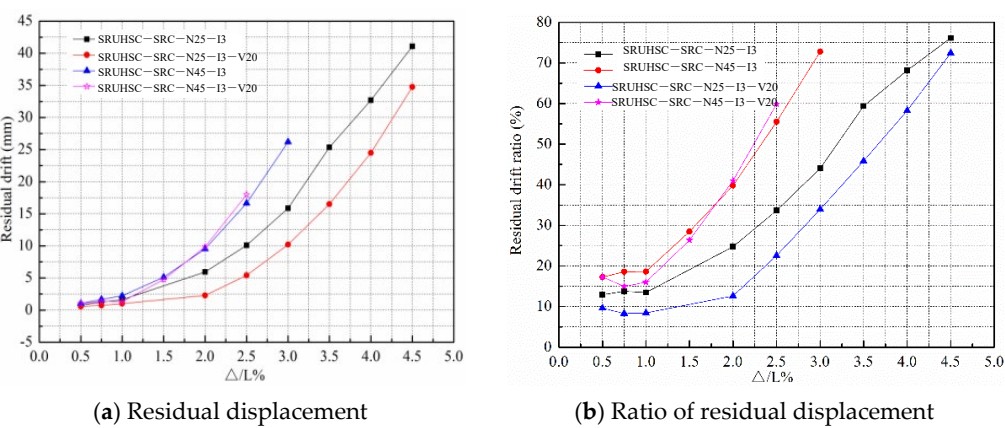

(**a**) Residual displacement      (**b**) Ratio of residual displacement

**Figure 10.** Residual deformation curves.

When the axial compression ratio is 0.45, the effect of rapid loading on the residual displacement and residual displacement ratio of the structure is similar to that of the pseudo-static test, whereas the deformation capacity of rapid loading is smaller than that of the pseudo-static test. It is indicated that under the action of a high axial compression ratio, the effect of the high loading rate on the overall structure is very weak.

Under conditions of rapid loading, the amplitude of the structure of residual deformation increases with the increase in the axial compression ratio, which is consistent with the quasi-static test. However, the variation in residual displacement and the residual displacement ratio under rapid loading is more evident than in the pseudo-static test.

### 3.7. Strain Analysis

The variations in longitudinal reinforcement, stirrups and steel bones in the structure can directly reflect the specific strain of each part of the whole structure in the process of deformation.

It can be seen from Figure 11 that the variation trends of the two were similar, and both underwent compressive deformations. After loading for a certain period of time, the reinforcement was in the plastic deformation stage, but the strain variation range of the quasi-static test was larger than that in the rapid loading test.

The stirrup strain of the quasi-static test was very small, as shown in Figure 11. In the process of loading, the stirrup strain value increased gradually at the beginning of loading. The strain value increased very quickly after a certain value in the stage of plastic deformation was reached. The stirrup strain had a reciprocating change and the strain value did not continue to increase at the end of the test, indicating that the degree of damage of the rapid-loading test was also more serious than that of the static test.

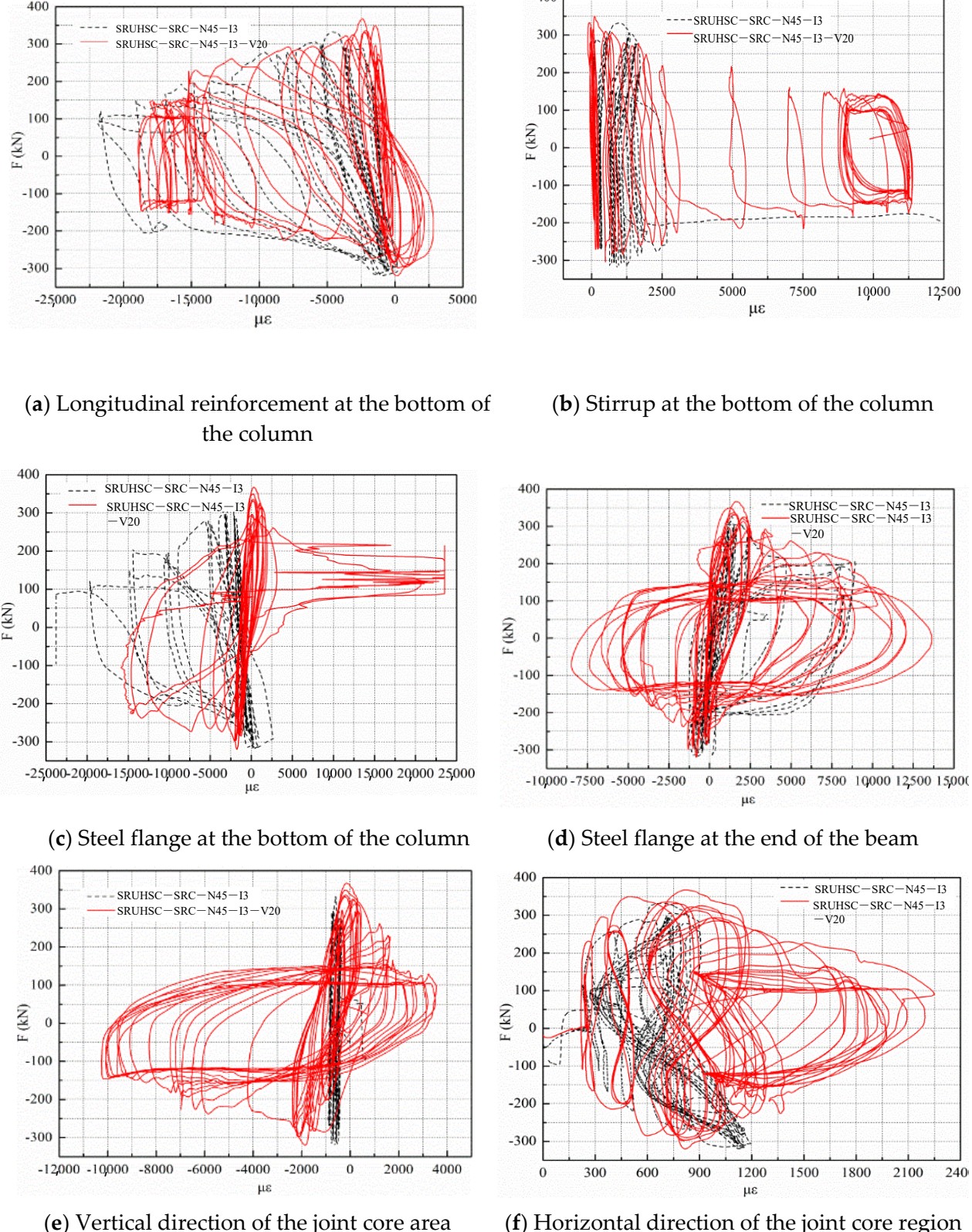

(**a**) Longitudinal reinforcement at the bottom of the column

(**b**) Stirrup at the bottom of the column

(**c**) Steel flange at the bottom of the column

(**d**) Steel flange at the end of the beam

(**e**) Vertical direction of the joint core area

(**f**) Horizontal direction of the joint core region

**Figure 11.** Strain analyses of the various areas.

The strains of steel bone flanges at the column's bottom and beam end are shown in Figure 11c,d. In the pseudo-static test, the strains at the column bottom tend to be of compressive deformation, while at the beam end, they tend to be of tensile deformation.

However, in the rapid-loading test, the tension and compression deformation of the steel bone flange were almost balanced, and no obvious buckling deformation occurred.

The shear strain of the steel bone web's vertical direction and horizontal direction are shown in Figure 11e and Figure 11f, respectively. The steel-reinforced web's vertical strain deformation was very small in the quasi-static loading experiment. While horizontal strain deformation occurred after a certain level of deformation, there was no obvious increase in reciprocating deformation stages. At the same time, the vertical and horizontal strain values changed obviously in the rapid-loading experiments. The vertical strain value was slightly larger than that in the static test. In the later loading stages, the strain hysteresis phenomenon appeared gradually and a trend of compression developed. The horizontal strain was mainly in the form of tensile deformation.

In conclusion, the internal structure of longitudinal reinforcement, stirrup and steel bone strain values were greater than in the pseudo-static test. This indicates that the change in the structure section and the degree of damage under the action of rapid loading were more serious than those in the static test.

## 4. Conclusions

In this paper, the seismic performance of steel-frame ultra-high-strength concrete frame structures under different loading rates (0.5 mm/s and 20 mm/s, respectively) are analyzed. Under high and low axial compression ratios, the effects of different loading rates on the seismic performance of structures and of different axial compression ratios on the seismic performance of structures under high loading rates are verified. The specific experimental results are as follows:

(1) The horizontal resistance capacity of structures can be significantly improved by rapid loading in the elastic and elastic–plastic ranges. The loading rate has a significant impact in those stages of structural stress. After reaching the plastic deformation point, the effect of the loading rate on the horizontal resistance capacity of the structure is weakened.

(2) Rapid loading can reduce the rate of horizontal resistance capacity declination and reduce the bond-slip behavior of the structure under the same amplitude in the plastic deformation stage, which has the effect of improving the stability of the structure and increasing the deformation capacity of the structure.

(3) The ductility coefficient of the structure increases with the same axial compression ratio under rapid loading. The ductility coefficient increases greatly with a low axial compression ratio, while the ductility coefficient increases slightly with a high axial compression ratio. This shows that the loading rate has a significant effect on the frame with a small axial compression ratio.

(4) With the increase in loading rate, the secant stiffness of the structure is improved. When the axial compression ratio is 0.25, the slope of stiffness degradation increases; when the axial compression ratio is 0.45, only the overall secant stiffness value of the structure increases and the effect on the slope of stiffness degradation is not obvious.

(5) The effect of rapid loading on the energy dissipation capacity of the structure is not obvious, while the energy dissipation capacity is slightly lower than that of the static test.

(6) The variation of residual displacement and the residual displacement ratio under rapid loading is larger than the pseudo-static test value. The effect of loading rate on the residual displacement and displacement ratio is obvious when the axial pressure is small. The residual deformation increases and the overall deformation capacity decreases with an increase in axial compression ratio.

**Author Contributions:** Conceptualization, W.L. and Y.M.; methodology, W.L.; software Y.M.; validation, W.L., Y.M. and J.J.; formal analysis, Y.M.; writing—original draft preparation, W.L.; writing—review and editing, W.L.; funding acquisition, W.L. and J.J. All authors have read and agreed to the published version of the manuscript.

**Funding:** This research was funded by the National Natural Science Foundation of China (Grant: 51178078), the Natural Science Foundation of the Science Department of Liaoning Province (Grant: 2020-BS-215) and the Scientific Research Program of the Education Department of Liaoning Province (Grant: LJKZ0721).

**Informed Consent Statement:** Informed consent was obtained from all subjects involved in the study.

**Data Availability Statement:** Data supporting the reported results can be found in publicly archived datasets analyzed during the study.

**Acknowledgments:** Acknowledge the State Key Laboratory of Coastal and Offshore Engineering for the experimental conditions.

**Conflicts of Interest:** The authors declare no conflict of interest.

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
