# Peer review of "Effect of Fast Loading on the Seismic Performance of SRUHSC Frame Structures"

_buildings, doi:10.3390/buildings12060736_

Round 1
Reviewer 1 Report
The submitted paper presents a limited amount of test results emphasizing on the influence of the loading rate on the cyclic of concrete frames.
We have a major problem with this paper: nothing is said about the test specimens. No drawings, sketches or pictures, no dimension, no material properties, nothing about the loading… As a consequence, the codes defining the tests in table 1 are impossible to understand.
Although the description of the results reads fine, it is difficult to evaluate them without knowing about the object of the investigation.
Therefore the authors are invited to add a section giving all details about the test specimens and on the loading (preloading, test set-up, application of the cyclic load…) in order to allow a correct evaluation of the results.
Author Response
Dear editor and reviewers:
Thank you very much for your letter and the comments from the referees about our paper 《Effect of Fast Loading on Seismic Performance of SRUHSC Frame Structure》(1706253).Your comments are of great significance to the improvement of the scientificity and innovation of the paper.We have checked the manuscript and revised it according to the comments. We submit here the revised manuscript as well as a list of changes. All changes are highlighted in the paper.
If you have any question about this paper,please don’t hesitate to let me know. Thank you.
Sincerely yours,
Dr. Wei Liu.
Response to reviewer 1:
Thank you very much for your comments on our paper. We have revised according to your comments.
Comments:The submitted paper presents a limited amount of test results emphasizing on the influence of the loading rate on the cyclic of concrete frames.
We have a major problem with this paper: nothing is said about the test specimens. No drawings, sketches or pictures, no dimension, no material properties, nothing about the loading… As a consequence, the codes defining the tests in table 1 are impossible to understand.
Although the description of the results reads fine, it is difficult to evaluate them without knowing about the object of the investigation.
Therefore the authors are invited to add a section giving all details about the test specimens and on the loading (preloading, test set-up, application of the cyclic load…) in order to allow a correct evaluation of the results.
Reply: We have added a section as 2.1 and 2.2 giving all details about the test specimens and on the loading according to your comments.

Reviewer 2 Report
Review for Buildings
Manuscript ID: buildings-1706253
Title: Effect of Fast Loading on Seismic Performance of SRUHSC Frame Structure
In the present work, the effect of axial compression ratio on the seismic performance of single-storey single-span steel frame structure with ultra-high strength concrete under rapid loading has been investigated by means of some experimental tests. The failure mode, deformation, strength and stiffness degradation, energy dissipation capacity and residual displacement of the ultra-high strength concrete frame structure are compared and analyzed by considering two different loading rates (fast and pseudo-static load) and two different axial compression ratios.
The subject addressed in this article is worthy of investigation and the paper is of sufficient technical quality. However major revisions are required to make the paper acceptable for publication in “Buildings”. To this end the authors are encouraged to prepare a revised version of the paper in which the following issues should be considered.
Comments:
- In Paragraph “2. Experimental program”, the authors should insert a new paragraph containing the geometry configuration of the performed tests, the mechanical properties of the materials and the type of the employed rebars.
- Figure 1 is not cited into the text of the manuscript.
- In paragraph 2.2 the authors explain the differences, in term of crack pattern, between the fast and pseudo-static test. Please the authors to highlight the essential parameters of these loading conditions (fast and pseudo-static) into paragraph 2.1.
- The parameter nt, reported at line 109 and 117, must be explained by the authors. I think you are referring to the axial compression ratio. Please insert this information in the new paragraph required by the Comment 1.
- The results, in terms of crack pattern, reported in Paragraph 2.2, should be enlarged considering for example, crack patterns at loading steps before the complete failure, if possible.
- In Paragraph 3.1, the authors should introduce into the text the label of performed test that are reported in the graph of Fig. 3. For example, what does the number 13 in the SRUHSC-SRC-N25-13 label mean?
- Please, in Paragraph 3.3 the authors should insert into the text the meaning of symbol µΔ and θu (ultimate displacement angle).
- The Introduction is short and limited with respect to the wide class of the investigated problem. The authors, for example, could explain the recent numerical and experimental studies about the high-strength and ultra-high-strength concrete materials subjected to flexural and compression stress states at the material level (as statement by the authors at Line 52). Eventually, the bibliographic context in the Introduction Section of the paper could be properly enlarged improving the literature review of the recent works. To this end, I recommend to discuss about these works:
- Experimental and numerical study on mechanical properties of Ultra High Performance Concrete (UHPC), Construction and Building Materials, Volume 156, 15 December 2017, Pages 402-411. https://doi.org/10.1016/j.conbuildmat.2017.08.170
- Failure analysis of ultra high-performance fiber-reinforced concrete structures enhanced with nanomaterials by using a diffuse cohesive interface approach, Nanomaterials 10 (9), 1792, http://dx.doi.org/10.3390/nano10091792
Author Response
Dear editor and reviewers:
Thank you very much for your letter and the comments from the referees about our paper 《Effect of Fast Loading on Seismic Performance of SRUHSC Frame Structure》(1706253).Your comments are of great significance to the improvement of the scientificity and innovation of the paper.We have checked the manuscript and revised it according to the comments. We submit here the revised manuscript as well as a list of changes. All changes are highlighted in the paper.
If you have any question about this paper,please don’t hesitate to let me know. Thank you.
Sincerely yours,
Dr. Wei Liu.
Response to reviewer 2:
- Comments:In Paragraph “2. Experimental program”, the authors should insert a new paragraph containing the geometry configuration of the performed tests, the mechanical properties of the materials and the type of the employed rebars.
Reply: We have inserted a new paragraph containing the geometry configuration of the performed tests, the mechanical properties of the materials and the type of the employed rebars as shown in Fig.1, Fig.2 and Tab.1-3.
- Comments:Figure 1 is not cited into the text of the manuscript.
Reply: We have revised according to your comments.
- Comments:In paragraph 2.2 the authors explain the differences, in term of crack pattern, between the fast and pseudo-static test. Please the authors to highlight the essential parameters of these loading conditions (fast and pseudo-static) into paragraph 2.1.
Reply: We have highlighted the essential parameters of these loading conditions according to your comments. The essential parameter of these loading conditions is the loading rate.The loading rates of this test are 0.5mm/s and 20mm/s respectively in pseudo-static and fast loading, and the corresponding material strain rates are 9.6×10-5 and 3.8×10-3respectively.
- Comments:The parameter nt, reported at line 109 and 117, must be explained by the authors. I think you are referring to the axial compression ratio. Please insert this information in the new paragraph required by the Comment 1.
Reply:We have inserted this information in 2.1.
- Comments:The results, in terms of crack pattern, reported in Paragraph 2.2, should be enlarged considering for example, crack patterns at loading steps before the complete failure, if possible.
Reply: We have revised in 2.3 according to your comments.
- Comments:In Paragraph 3.1, the authors should introduce into the text the label of performed test that are reported in the graph of Fig.3. For example, what does the number 13 in the SRUHSC-SRC-N25-13 label mean?
Reply:We have added this information in Tab.1.
7.Comments:Please, in Paragraph 3.3 the authors should insert into the text the meaning of symbol µΔ and θu (ultimate displacement angle).
Reply:We have added this information in 3.3.
- Comments:The Introduction is short and limited with respect to the wide class of the investigated problem. The authors, for example, could explain the recent numerical and experimental studies about the high-strength and ultra-high-strength concrete materials subjected to flexural and compression stress states at the material level (as statement by the authors at Line 52). Eventually, the bibliographic context in the Introduction Section of the paper could be properly enlarged improving the literature review of the recent works. To this end, I recommend to discuss about these works:
Experimental and numerical study on mechanical properties of Ultra High Performance Concrete (UHPC), Construction and Building Materials, Volume 156, 15 December 2017, Pages 402-411. https://doi.org/10.1016/j.conbuildmat.2017.08.170
Failure analysis of ultra high-performance fiber-reinforced concrete structures enhanced with nanomaterials by using a diffuse cohesive interface approach, Nanomaterials 10 (9), 1792, http://dx.doi.org/10.3390/nano10091792
Reply: We have enlarged improving the literature review of the recent works in introduction.

Reviewer 3 Report
The article titled “Effect of Fast Loading on Seismic Performance of SRUHSC Frame Structure” presents a novel and interesting study. Very few studies have been conducted on the behavior of SRUHSC under seismic/fast loadings. Furthermore, the study is according to the scope of the journal and deserves a publication after revision.
- In my opinion the weakest part of this article is the use of English language. It needs extensive revision before publication.
- SRUHSC is short for steel-reinforced ultra-high strength concrete. Not steel reinforcement.
- Line 10: “structures were used” should be instead of “structure was used”. The writing should be simple without disruption of flow.
- I would recommend that key conclusions should be added as the list 2-3 lines in the abstract.
- Keywords should be revised throughout. The words which have been used in title should not be used in the keywords. Instead, other important words which you missed in the title should be used as keywords to increase the visibility of your article.
- Make sure the reference style is as per Buildings journal format.
- Introduction needs more background information and literature.
- I appreciate the outstanding efforts of authors in the results and discussion part.
- Conclusion should be revised to make it more concise and crispier.
Author Response
v
Dear editor and reviewers:
Thank you very much for your letter and the comments from the referees about our paper 《Effect of Fast Loading on Seismic Performance of SRUHSC Frame Structure》(1706253).Your comments are of great significance to the improvement of the scientificity and innovation of the paper.We have checked the manuscript and revised it according to the comments. We submit here the revised manuscript as well as a list of changes. All changes are highlighted in the paper.
If you have any question about this paper,please don’t hesitate to let me know. Thank you.
Sincerely yours,
Dr. Wei Liu.
Response to reviewer 3:
Thank you very much for your comments on our paper. We have revised according to your comments.
- Comments: In my opinion the weakest part of this article is the use of English language. It needs extensive revision before publication.
Reply: We have revised the language according to your comments.
- Comments:SRUHSC is short for steel-reinforced ultra-high strength concrete. Not steel reinforcement.
Reply: We have revised according to your comments.
- Comments:Line 10: “structures were used” should be instead of “structure was used”. The writing should be simple without disruption of flow.
Reply: We have revised according to your comments.
- Comments:I would recommend that key conclusions should be added as the list 2-3 lines in the abstract.
Reply:We have revised the abstract according to your comments.
- Comments:Keywords should be revised throughout. The words which have been used in title should not be used in the keywords. Instead, other important words which you missed in the title should be used as keywords to increase the visibility of your article.
Reply:We have revised the keywords according to your comments.
- Comments:Make sure the reference style is as per Buildings journal format.
Reply: We have checked and revised the reference according to your comments.
- Comments:Introduction needs more background information and literature.
Reply:We have added more background information and literature according to your comments.
- Comments:Conclusion should be revised to make it more concise and crispier.
Reply: We have revised according to your comments.

Round 2
Reviewer 1 Report
Thanks for the improvements brought in the updated version. In our opinion, the paper is acceptable in its current version.
Reviewer 2 Report
The revised manuscript can be published in Buildings in present form.